# Bitiscetin-3, a Novel C-Type Lectin-like Protein Cloned from the Venom Gland of the Viper *Bitis arietans*, Induces Platelet Agglutination and Inhibits Binding of Von Willebrand Factor to Collagen

**DOI:** 10.3390/toxins14040236

**Published:** 2022-03-25

**Authors:** Youhei Nashimoto, Fumio Matsushita, Johannes M. Dijkstra, Yuta Nakamura, Hidehiko Akiyama, Jiharu Hamako, Takashi Morita, Satohiko Araki, Taei Matsui

**Affiliations:** 1Graduate School of Medical Sciences, Fujita Health University, 1-98 Kutsukake-cho, Toyoake 470-1192, Japan; f41005055yn@hotmail.co.jp (Y.N.); fumio@fujita-hu.ac.jp (F.M.); yuta2019453@gmail.com (Y.N.); hakiyama@fujita-hu.ac.jp (H.A.); 2Institute for Comprehensive Medical Science, Fujita Health University, 1-98 Kutsukake-cho, Toyoake 470-1192, Japan; dijkstra@fujita-hu.ac.jp; 3Faculty of Medical Management and Information Science, Fujita Health University School of Health Sciences, 1-98 Kutsukake-cho, Toyoake 470-1192, Japan; jhamako@fujita-hu.ac.jp; 4Department of Biochemistry, Meiji Pharmaceutical University, 2-522-1 Noshio-cho, Kyose 204-8588, Japan; tmorita@my-pharm.ac.jp; 5Graduate School of Science, Nagoya University, Furo-cho, Chikusa-ku, Nagoya 464-8602, Japan; saraki@bio.nagoya-u.ac.jp

**Keywords:** bitiscetin, von Willebrand factor, platelet agglutination, snake venom, GPIb, collagen, *bitis arietans*, c-type lectin, snaclec

## Abstract

Bitiscetin-1 (aka bitiscetin) and bitiscetin-2 are C-type lectin-like proteins purified from the venom of *Bitis arietans* (puff adder). They bind to von Willebrand factor (VWF) and—at least bitiscetin-1—induce platelet agglutination via enhancement of VWF binding to platelet glycoprotein Ib (GPIb). Bitiscetin-1 and -2 bind the VWF A1 and A3 domains, respectively. The A3 domain includes the major site of VWF for binding collagen, explaining why bitiscetin-2 blocks VWF-to-collagen binding. In the present study, sequences for a novel bitiscetin protein—bitiscetin-3—were identified in cDNA constructed from the *B. arietans* venom gland. The deduced amino acid sequences of bitiscetin-3 subunits α and β share 79 and 80% identity with those of bitiscetin-1, respectively. Expression vectors for bitiscetin-3α and -3β were co-transfected to 293T cells, producing the heterodimer protein recombinant bitiscetin-3 (rBit-3). Functionally, purified rBit-3 (1) induced platelet agglutination involving VWF and GPIb, (2) did not compete with bitiscetin-1 for binding to VWF, (3) blocked VWF-to-collagen binding, and (4) lost its platelet agglutination inducing ability in the presence of an anti-VWF monoclonal antibody that blocked VWF-to-collagen binding. These combined results suggest that bitiscetin-3 binds to the A3 domain, as does bitiscetin-2. Except for a small N-terminal fragment of a single subunit—which differs from that of both bitiscetin-3 subunits—the sequences of bitiscetin-2 have never been determined. Therefore, by identifying and analyzing bitiscetin-3, the present study is the first to present the full-length α- and β-subunit sequences and recombinant expression of a bitiscetin-family toxin that blocks the binding of VWF to collagen.

## 1. Introduction

Toxic proteins in snake venoms can generally be classified as neurotoxic, dermonecrotic, proteolytic, hemotoxic, and ion channel inhibitor toxins. They form useful tools for exploring or controlling neuronal transduction [1], hemostasis and the coagulation cascade [2], hemagglutination [3], blood pressure [4,5,6], and platelet aggregation [7,8,9,10,11], while also contributing to drug discovery [12]. A group of snake venom proteins belong to a family of domain-swapped heterodimers of single C-type lectin (CTL)-like domain molecules (“Snaclecs”) [13,14,15]. Snaclecs are mostly heterodimers of covalently-linked α and β subunits and show anticoagulant or coagulant, and platelet aggregation agonist or antagonist activities. Well-known among these are the snake venom CTL-like proteins botrocetin and bitiscetin-1 (aka bitiscetin), derived from *Bothrops jararaca* (jararaca, a South-American viper) and *Bitis arietans* (puff adder, an African viper), respectively. Botrocetin and bitiscetin-1 induce platelet agglutination by binding and modulation of the von Willebrand factor (VWF) [16,17,18]. VWF is an adhesion molecule that attaches flowing platelets to subendothelial matrices such as collagen at the vascular injury [19,20]. This platelet-plug formation is essential for the early stage of hemostasis. Single VWF molecules of 270 kDa form the subunits of disulfide-linked multimeric VWF structures. A single VWF molecule comprises sets of similar domains in the order (with the same letter designations for similar domains): _2_HN-D’-D3-A1-A2-A3-D4-C1-C2-C3-C4-C5-C6-CN-COOH. The A1, A2, and A3 domains confer different functions and, respectively, contain binding sites for platelet glycoprotein Ib (GPIb), the cleaving site for ADAMTS13, and the major binding site for collagen [19]. The toxins botrocetin and bitiscetin-1 each bind to the A1 domain of VWF, after which the toxin-A1 complex can stably bind platelet GPIb and thereby induce platelet agglutination [21,22]. Botrocetin and bitiscetin-1 are useful diagnostic reagents for hemostatic disorders such as von Willebrand disease [23,24].

The binding sites on VWF A1 for bitiscetin-1 and botrocetin partially overlap, although their angles of binding are very different and almost orthogonal to each other [9,25,26]. Initially, it was speculated that botrocetin and bitiscetin-1 induce conformational changes in the A1 domain, leading to A1 binding of GPIb. However, the comparison between the crystal structures of bitiscetin-1-A1 or botrocetin-A1 complex and that of toxin-free A1 could not detect significant differences in A1 conformation [25]. Finally, the crystal structure of a botrocetin-A1-GPIb ternary complex was determined, showing that botrocetin itself provides an extra binding surface for GPIb; this is now believed to cause a higher GPIb-affinity of botrocetin-A1 compared to A1 alone and to explain how botrocetin induces platelet agglutination [26]. Based on modeling, a similar mode of action has been proposed for bitiscetin-1 [22], but a bitiscetin-1-A1-GPIb structure has not been elucidated yet.

After our first isolation and characterization of bitiscetin-1 from the venom of *B. arietans* [18] by similar means and using the venom of the same species, Obert et al. [27] isolated a similar protein which they originally thought to be the same bitiscetin despite showing some differences. A later analysis by the same group, which included the sequencing of 12 amino acids of the N-terminus of a single subunit, provided definite evidence that this was a different protein which then was coined “bitiscetin-2” [28]. Bitiscetin-2 purified from *B. arietans* venom induced GPIb-dependent binding of VWF to platelets [27]. However, unlike bitiscetin-1, it binds to the VWF A3 domain which contains the major collagen-binding site, and an affinity for the VWF A1 domain could not be detected [27]. Nevertheless, analysis using monoclonal antibodies revealed that bitiscetin-2 induced conformational changes in the VWF A1 domain, which was postulated as the mechanism of how bitiscetin-2 induces the binding of VWF to platelets; this binding could be reverted by anti-GPIb monoclonals, and the combined results suggest that bitiscetin-2 can induce platelet agglutination in a VWF-GPIb dependent manner [28]. However, a platelet agglutinating effect of bitiscetin-2 has not been reported. Moreover, despite its unique properties, a sequence analysis of bitiscetin-2 has never been completed, interfering with the understanding of this protein and its possible use as a recombinant biomedical tool.

In the present study, we report the cDNA cloning and recombinant expression of a novel bitiscetin-family protein from *B. arietans* that we designated bitiscetin-3. Its sequences are different from those of bitiscetin-1 and -2. We directly compared the functions of bitiscetin-1 and -3 and found them to be quite different. Bitiscetin-3 induces VWF- and GPIb-involving platelet agglutination, but inhibits VWF-to-collagen binding and probably binds to the VWF A3 domain, which is reminiscent of previous findings for bitiscetin-2. This is the first report showing full sequences of a bitiscetin-family toxin with such properties.

## 2. Materials and Methods

### 2.1. Materials

The venom gland of *Bitis arietans*, which was eviscerated and immediately frozen in liquid nitrogen, was purchased from The Japan Snake Institute (Ota, Japan). Bitiscetin-1 and botrocetin were purified from the crude venoms of *B. arietans* and *Bothrops jararaca* (Sigma-Aldrich, St. Louis, MO, USA or Latoxan, Rosans, France), respectively, as previously described [17,18]. VWF was purified from factor VIII concentrates as previously described [29]. Human 293T cells were purchased from the RIKEN Bioresource Center (Tsukuba, Japan). The monoclonal antibodies (mAbs) HIP1 against GPIbα and NMC-4 against the VWF A1 domain [30] were from EXBIO (Vestec, Czech) and a generous gift from Dr. Y. Fujimura of Nara Medical University, respectively. The mAbs directed against VWF (VW28-1, 33-5, 40-1, 52-3 and 93-2) were from Takara Bio (Otsu, Japan). Other reagents described in the article were biochemical grade or molecular biological grade and purchased from Wako chemicals (Osaka, Japan) or Sigma.

### 2.2. Construction of cDNA and a Library of cDNA-Containing Vectors

Total RNA was extracted from the *B. arietans* venom gland (1 g) using TRIZOL (Invitrogen, Carlsbad, CA, USA), and mRNA was isolated from total RNA by a Poly(A)Purist mRNA purification kit (Ambion, Austin, TX, USA). cDNA was synthesized by the SuperScript Choice system (Invitrogen) using an oligo-dT primer. This cDNA was used for bitiscetin-specific PCR experiments. In addition, the cDNA was also cloned into Lambda ZAP II vector (Stratagene, La Jolla, CA, USA) to create a library, and the amplified library was used to find the open reading frame ends. For all procedures we followed the instructions provided by the kit supplier.

### 2.3. Primers

For the initial bitiscetin-specific PCR search using cDNA, a degenerate forward primer Snaclec-deg-F, based on signal sequence coding fragments of snaclec genes in *B. arietans*, 5′-ATGGGGCGATTCATCTTCVTCAGC (V = A, C, or G), was used in conjunction with a degenerate reverse primer based on a conserved part of the alpha or beta chains. For alpha, this primer was Bit-α-deg-R, 5′-CANGGNARRTCNGTCCA (R = A or G; N is A, C, G, or T), and for beta this was Bit-β-deg-R, 5′-CANGTCCAYTGDATCCA (Y = C or T; D = A, G, or T).

For PCR amplification of fragments including *bitiscetin-3* open reading frame ends from the Lamda ZAP II vector cDNA library, a bitiscetin-specific primer was used in conjunction with the vector specific primers SK-primer, 5′-GCCGCTCTAGAACTAGTGGATCC, or KS-primer, 5′-CCTCGAGGTCGACGGTATCG. The bitiscetin-specific primers in these assays were: Bit-α-F1, 5′-TGGGAAGATGCAGAGAAATTCTGCGTGGAGAACA; Bit-β-F1, 5′-TGGGCCGATGCAGAGAAATTCTGCAAGGAGCT; Bit-α-R1, 5′-CCACTGTGGGCTGCATTGCTGTGTTTTGCT; or Bit-β-R1, 5′-GCCTGGCACCATCAGTCCACCTCAATGGGCAAAT.

For the PCR amplification of full-length *bitiscetin-3α* and *bitiscetin-3β* open reading frames from cDNA, while adding restriction enzyme cloning sites and a Kozak motif for efficient translation, the following primer pairs were used, respectively: Bit-α-F2, 5′-GAATTCACCATGGGGCGATTCATCTTCC + Bit-α-R2, 5′-GCGGCCGCCTGGATCTTAATGCGGTAG, and Bit-β-F2, 5′-GAATTCACCATGGGGCGATTCATCTTCC + Bit-β-R2, 5′-GCGGCCGCGATTTCAATTTGGTACCC.

For PCR experiments with primers specifically recognizing *bitiscitin-1β* or *-3β*, two forward and two reverse primers were used per gene in the combinations F1 + R1 and F2 + R2; all primers had similar Tm values. For *bitiscitin-1β*, these primers were: Bit-1β-F1, 5′-GGCATTGCTACAAGGTCTTCAAAGTAGAG; Bit-1β-F2, 5′-GTGAACGGTGGGCATCTCATG; Bit-1β-R1, 5′-TACCCGGTACTTGCAGACAAAAGACT; and Bit-1β-R2, 5′-GCAGGTCCATTGAATCCATTTGTTATG. For *bitiscitin-3β*, these primers were: Bit-3β-F1: 5′-GGCATTGCTACAAGGTCTTCAAGGAGT; Bit-3β-F2: 5′-GAACGGCGGGCATCTCAC; Bit-3β-R1, 5′-CGGTACTTGCAGACGAAACCCAGTAC; and Bit-3β-R2, 5′-CTGCACTTCCATTGAAACCATTTGTTATC.

### 2.4. PCR and Sequencing

PCR with degenerate primers using cDNA, or PCR with a bitiscetin-specific primer plus a vector-specific primer using the Lamda ZAP II vector cDNA library, was performed using Go Taq Green Master Mix (Promega, Madison, WI, USA) with 35 cycles in which the annealing temperature differed per primer set (denaturation at 94 °C for 10 s, annealing at 40–55 °C for 30 s, and elongation at 72 °C for 60 s). PCR products were isolated from agarose gel and ligated into pGEM-T easy vector (Promega), after which the plasmids were transformed to *E. coli* for propagation. For multiple individual clones (>10) per PCR product, plasmids were purified by QIAprep Spin Miniprep Kit (Qiagen, Germantown, MD, USA) and sequenced using Big Dye Terminator v3.1 Cycle Sequencing Kit (Applied Biosystems, Waltham, MA, USA) with ABI PRISM 3100-Avant Genetic Analyzer (Applied Biosystems).

For PCR amplification from cDNA of full-length *bitiscetin-3α* and *-3β* while adding EcoRI (forward) and NotI (reverse) restriction sites, touchdown PCR was performed using the PrimeSTAR HS Taq polymerase kit (Takara) and 30 cycles (denaturation at 94 °C for 10 s, annealing and elongation from at 75 °C for 60 s with 1 °C decrease every cycle until 65 °C, and final annealing and elongation at 64 °C for 60 s). PCR products were purified, digested with EcoRI and NotI, and cloned into the expression vector pCAGEN (Addgene, Watertown, MA, USA). Plasmids were transformed to *E. coli* and multiple individual clones were analyzed as described above. For PCR reactions aimed to distinguish between *bitiscetin-1* and *bitiscetin-3* using cDNA, Takara emerald Taq (Takara) was used with 30 cycles (94 °C 10 s, 60 °C 30 s, 72 °C 30 s). For all PCR and sequencing reactions, the protocols provided by the kit suppliers were followed.

### 2.5. GenBank Accession Numbers of Bitiscetin-3α and -β

The sequences of *bitiscetin-3α* and *-β* were deposited as GenBank accessions LC500203 and LC500204.

### 2.6. Sequence Comparisons

Deduced amino acid sequences of bitiscetin-3 subunits were compared against the NCBI sequence database using the program BLAST [31]. For making the sequence alignment figure, the sequences were aligned by hand, based on considerations of similarity and evolution [32]. The open reading frames of bitiscetin-1 (KU724120) and bitiscetin-3 were compared by the Synonymous Non-synonymous Analysis Program, SNAP v2.1.1, https://www.hiv.lanl.gov/content/sequence/SNAP/SNAP.html (accessed on 10 January 2022) [33]. A phylogenetic tree was made by Neighbor-Joining method using MEGA7 software [34].

### 2.7. Recombinant Expression of cDNA

The 293T cells were grown in DMEM (Gibco, Tokyo, Japan) supplemented with 10% fetal calf serum, and the medium was changed to OPTI-MEM (Gibco) 2 h before transfection. Cells were transfected with the same amount of pCAGEN-bitiscetin-3α and pCAGEN-bitiscetin-3β using GeneJuice (Novagen, Madison, WI, USA), in accordance with the manufacturer’s instructions. After 48–72 h, the cultured medium was collected and centrifuged at 3000 rpm for 10 min to remove cells and cell debris, and then the supernatant was concentrated and substituted to TBS (150 mM NaCl containing 10 mM Tris-HCl, pH 7.5) using an Amicon Ultra-15 (10 K) filter (Millipore, Billerica, MA, USA), after which the sample was frozen until use.

### 2.8. Purification of Recombinant Protein

The concentrated medium was subjected to gel filtration on a Superdex75 column using an AKTApurifier 10 system (GE Healthcare, Tokyo, Japan) at 4 °C. Fractions showing platelet agglutinating activity with VWF using formalin-fixed platelets were collected and concentrated with Amicon Ultra (Millipore), substituting medium with 20 mM sodium acetate buffer, pH 5.0. The active fraction was further purified by cation-exchange chromatography on a MonoS column using an increasing gradient of NaCl. Peak fractions were examined for platelet agglutinating activity and active fractions were concentrated. The purity was evaluated by SDS-polyacrylamide gel electrophoresis (SDS-PAGE), and the protein concentration was determined by the BCA assay method (Pierce, Rockford, IL, USA) using bovine serum albumin as a standard.

For N-terminal amino acid sequence analysis, reduced proteins were transferred to a poly-vinylidene difluoride (PVDF) membrane (Millipore). After staining with Coomassie Brilliant Blue G (Sigma-Aldrich), bands were sliced and subjected to direct sequencing using ABI Procise 494 protein sequencer as previously described [35].

### 2.9. Platelet Agglutination Assay

After obtaining informed consent, blood was drawn from healthy donors into a tube containing 1/10 volume of 3.2% sodium citrate. Citrated blood was centrifuged at 250× *g* for 10 min and platelet-rich plasma (PRP) was collected. The remaining blood was centrifuged at 2500 rpm for 15 min in order to get platelet-poor plasma (PPP). Platelet suspension (3 × 10^8^/mL platelets, 250 µL) of PRP was stirred in a small cuvette at 37 °C for 2 min on an aggregometer (Mebanics model PT-2, Yokohama, Japan). Light transmission of PRP after the addition of 1–10 µL of test solution was monitored using PPP (for PRP) or TBS (for washed platelets) as a reference. Washed platelets were prepared from PRP as described [36]. TBS or anti-VWF monoclonal antibody solution was added (final concentration of 10 or 20 µg/mL) to PRP at 2–3 min before the addition of rBit-3 solution (2 µg/mL). In some experiments, collagen (20 µg/mL) or bitiscetin-1 (1 µg/mL) was added to the PRP. Washed platelets were preincubated with or without anti-GPIb monoclonal antibody HIP1 solution (10 µg/mL) for 3 min, and rBit-3 (1 µg/mL) and VWF (8 µg/mL) were successively added. Experiments were performed in duplo, with independent samples from two different donors; for each sample similar results were obtained, and representative results are shown. Formalin-fixed platelets were a gift from Dr. Matsumoto (Nara Medical University).

### 2.10. Western Blotting

After SDS-PAGE (17.5%), proteins on gels were electrically transferred onto PVDF membranes, and the membranes were incubated with anti-bitiscetin-1 mAbs, followed by detection with HRP-conjugated anti-mouse IgG (Zymed Lab. Inc., San Francisco, CA, USA) as described previously [22].

### 2.11. Enzyme-Linked Immuno-Sorbent Assay (ELISA) for Analyzing the Effect of rBit-3 on Binding of GPIb to VWF

Glycocalicin (the extracellular domain of GPIb) was prepared from platelets and biotinylated glycocalicin (B-GC) was prepared as described [22]. VWF solution (8 µg/mL) in TBS containing 1% BSA (B-TBS) was incubated in an ELISA plate (Nunc, Kamstrup, Denmark) precoated with anti-VWF (Dako, Kyoto, Japan) solution (10 µg/mL) for 90 min at 25 °C. After washing with TBS containing 0.05% tween-20 (T-TBS), immobilized VWF was incubated with B-GC (0–4 µg/mL in B-TBS) with or without bitiscetin-1 or rBit-3 (2 µg/mL) for 60 min. The binding of B-GC was detected by HRP-conjugated streptavidin (1:1000 diluted in B-TBS, Vector laboratory, Burlingame, CA, USA) and SigmaFast OPD reagent (Sigma-Aldrich, St. Louis, MO, USA) as described [37]. Absorbance at 490 nm was measured with a plate reader (Bio-Rad Model 680, Hercules, CA, USA).

### 2.12. ELISA for Analyzing the Effect of rBit-3 on Binding of Bitiscetin-1 to VWF

For some experiments, bitiscetin-1 was conjugated with biotin as described [17]. For a competition assay between bitiscetin-1 and rBit-3, biotinylated bitiscetin-1 (B-Bit-1: 0.04 µg/mL) was incubated in a VWF-captured ELISA plate as described above, in the presence of various concentrations (0–4 µg/mL) of rBit-3. The binding of B-Bit-1 was measured with HRP-conjugated streptavidin as described above.

### 2.13. ELISA for Evaluating the Effect of rBit-3 on VWF-to-Collagen Binding

Collagen type III solution (3 mg/mL, Nitta-Gelatin, Osaka, Japan) was diluted to 10 µg/mL in TBS and the collagen solution (50 µL) was coated to ELISA plates (Nunc) at 4 °C overnight. After washing with TBS, plates were blocked with B-TBS at 4 °C overnight. VWF solution (1 µg/mL in B-TBS) was mixed with the same volume of B-TBS solution containing either rBit-3 (0–40 µg/mL), bitiscetin-1 (0–40 µg/mL), or anti-VWF mAb (10 µg/mL) at 25 °C for 20 min, then added to the collagen-coated plate. The plates were incubated for 1 h, washed with T-TBS, and incubated with HRP-conjugated anti-VWF (Dako, Glostrup, Denmark) solution (diluted to 1:6000 in B-TBS) for 1 h. After washing with T-TBS, plates were incubated with SigmaFast OPD reagent for 5 min and the absorbance at 490 nm was measured as described above.

### 2.14. Statistical Analysis

Statistical analysis (unpaired *t* test) was performed, and graphs were obtained using GraphPad Prism v.7 (San Diego, CA, USA), and a *p* value < 0.03 was considered statistically significant.

## 3. Results

### 3.1. cDNA Cloning of a Novel Bitiscetin

From a venom gland of a *B. arietans* individual, cDNA was constructed. The initial screens for bitiscetin sequences with degenerate primers included a forward primer that matched the highly conserved signal sequence of *B. arietans* Snaclec proteins (Figure 1). This primer was used in conjunction with a degenerate reverse primer based on conserved amino acid sequence fragments of either bitiscetin α or β, namely WTDLPC and WIQWTC, respectively. Sequence analysis of amplified PCR fragments detected only a single type of sequence for each of the α and β subunit, despite analysis of multiple (>10) clones per cloned PCR fragment. Then, based on the obtained sequences, gene-specific primers were used in conjunction with a vector-specific primer to PCR amplify the gene open reading frame ends from a cDNA vector library. Eventually, using gene-specific primers, the full-length open reading frames of the new sequences, *bitiscetin-3α* and *-3β*, were amplified and determined. Sequences other than *bitiscetin-3α* and *-3β* were not detected for any of the PCR reactions. The bitiscetin-3α and -3β open reading frame sequences and their encoded amino acid sequences are shown in Figure 2.

### 3.2. Bitiscetin-1β cDNA Could Not Be Detected

The *B. arietans* cDNA sample was investigated by PCRs specific for either *bitiscetin-1β* or *bitiscetin-3β*. Figure 3 shows that both PCR reactions for *bitiscetin-3β* amplified a fragment of the expected size, whereas none of the two PCR reactions for *bitiscetin-1β* did. These results argue against the presence (of detectable amounts) of *bitiscetin-1β*. For *bitiscetin-1α*, the nucleotide sequence is not known, so we could not perform a similar experiment for that gene.

### 3.3. The Bitiscetin-3α and -3β Amino Acid Sequences

The deduced amino acid sequences of the bitiscetin-3 α and β subunits each contain the seven Cys residues that are typical for Snaclec proteins, including the cysteine that connects the subunits by disulfide bridge and participates in domain swapping between the subunits (Figure 4A) [14,38]. Upon comparison with protein sequences deposited in the NCBI database, both subunits showed the highest similarity with the respective bitiscetin-1 subunits, namely 79% and 80% amino acid, respectively. This is also reflected in the phylogenetic tree analysis result (Figure 4B). Compared to bitiscetin-1, each bitiscetin-3 subunit shows two extra amino acid residues and 27 residue replacements (Figure 4A).

Several, but not all, of the bitiscetin-1 residues that bind (defined as within 4.0 Å distance) to VWF are conserved in bitiscetin-3 (Figure 4A, light green and cyan shading for binding residues in the α- and β-subunit, respectively). Of the 18 bitiscetin-1 residues (five in the α subunit, 13 in the β subunit) involved in binding of VWF A1, only six are conserved in bitiscetin-3. This six out of 18 (33%) is considerably lower than the ~80% amino acid identities found for the full-length sequences, arguing against a conservation of VWF binding modes between bitiscetin-1 and -3. The N-termini of both bitiscetin-3α and -3β are different from the only twelve amino acid residues previously reported for bitiscetin-2 (Figure 4A) [28].

### 3.4. Recombinant Expression of Bitiscetin-3

Separate vectors for expressing the bitiscetin-3 α and β subunits were co-transfected to 293T cells, and the cultured medium was concentrated. The concentrates showed a platelet agglutination inducing activity, which was used as an assay tool for monitoring the presence of recombinant bitiscetin-3 protein (rBit-3) after purification into fractions by conventional methods. Through separation by gel-filtration and anion-exchange chromatography, a highly pure preparation of rBit-3 was obtained (Figure 5A). It showed a single band of ~25 kDa under non-reducing conditions and two similar-sized bands of ~15 kDa under reducing conditions (Figure 5A). In contrast, the ~25 kDa bitiscetin-1 heterodimer separated into two bands with clearly different sizes when under reducing conditions, namely ~17 kDa for the α-subunit and ~14 kDa for the β-subunit (Figure 5A). N-terminal amino acid sequencing was performed for each of the two bitiscetin-3 bands after their separation under reducing conditions. As the first 13 residues signals of the upper and lower bands were DEGXLPDXSXRVE and DEGXLPDXSSREG (X indicates unknown residues), they were identified as the α and β subunit of bitiscetin-3, respectively. The amino acid sequencing result confirmed that rBit-3 is a disulfide-linked heterodimer composed of α and β subunits. By comparison of the sequence results obtained by amino acid sequencing (of the mature peptides) and cDNA sequencing (of the full-length coding sequences), the signal sequences could be determined (Figure 2).

Binding specificities of anti-bitiscetin-1 mAbs for rBit-3 were assessed. One of the mAbs (ABIS-1) directed against the bitiscetin-1 β subunit showed a weak but clearly detectable reaction with the rBit-3 β subunit. Among the other antibodies recognizing the bitiscetin-1 α subunit (ABIS-5 and -8) or β subunit (ABIS-4), only ABIS-4 showed a weak reaction with rBit-3 (Figure 5B).

### 3.5. rBit-3 Induces Platelet Agglutination in Platelet-Rich Plasma, Which Can Be Blocked by Anti-VWF mAb NMC-4

rBit-3 induced platelet agglutination in platelet-rich plasma (PRP), although its activity was weaker than that of bitiscetin-1 (Figure 6A). Bitiscetin-1 was effective at 0.5 µg/mL, whereas rBit-3 at this concentration had little effect on PRP. Only with 2 µg/mL of rBit-3 was a pronounced platelet agglutination observed. This activity was inhibited in the presence of anti-VWF mAb NMC-4 (Figure 6B); the later addition of collagen as a positive control showed the ability of these platelets to still aggregate independent of VWF and GPIb [42] (Figure 6B). Because NMC-4 binds to the A1 domain of VWF and blocks its binding to GPIb [30], the results in Figs. 6A and 6B suggest that the platelet agglutination induced by rBit-3 involves VWF and GPIb.

### 3.6. rBit-3 Can Induce Agglutination of Washed Platelets, but Does So More Efficiently after the Addition of VWF

When rBit-3 was added to washed platelets (the washing removed VWF), a low level of platelet agglutination was observed, which was enhanced by adding VWF (Figure 6C). The agglutination of washed platelets could be inhibited by their prior incubation with anti-GPIb mAb HIP1 (Figure 6C). The result suggests that rBit-3 may directly interact with GPIb and so induce a weak agglutination, even in the absence of VWF.

### 3.7. rBit-3 Enhances Binding between GPIb and VWF, but Not as Efficient as Bitiscetin-1 or Botrocetin

The binding between GPIb and VWF in the presence of rBit-3 was evaluated using biotin-labeled glycocalicin (B-GC; glycocalicin is the extracellular domain of GPIb) and an ELISA-plate with immobilized VWF. When comparing the effects on B-GC binding of co-incubation with 2 µg/mL bitiscetin-1, botrocetin, or rBit-3, the enhancing effect of rBit-3 was much weaker than when using bitiscetin-1 or botrocetin (Figure 7A).

### 3.8. rBit-3 Does Not Compete with Bitiscetin-1 for Binding to VWF

For investigation of whether rBit-3 binds the same site on VWF as bitiscetin-1, a competition assay was performed. The binding of biotin-labelled bitiscetin-1 (B-Bit-1) to immobilized VWF was competitively inhibited by increasing amounts of non-labelled bitiscetin-1 (Figure 7B). However, even a high concentration of rBit-3 did not interfere with the binding of B-Bit-1 to VWF (Figure 7B). This proves that bitiscetin-3 does not bind to the same site on VWF as bitiscetin-1.

### 3.9. Effects of rBit-3 on the Collagen-VWF Interaction

VWF is known to bind very well to immobilized collagen type III [43]. This binding was reduced by rBit-3 in a concentration-dependent manner and could be entirely blocked by high rBit-3 concentrations (Figure 8A). In contrast, bitiscetin-1 had no effect on this binding (Figure 8A), which is in agreement with our previous report [22]. Furthermore, also consistent with our previous studies [43], among six investigated anti-VWF mAbs, only VW53-2 clearly inhibited the binding of VWF to collagen (Figure 8B). Pretreatment with VW53-2 prohibited rBit-3-induced platelet agglutination but had no significant effect on the platelet agglutination induced by bitiscetin-1 (Figure 8C). This suggests that rBit-3 binds to the major site of VWF for binding collagen type III, known to locate in the VWF A3 domain [44].

## 4. Discussion

In the present study, we have identified transcript sequences of a novel bitiscetin-family protein—bitiscetin-3—in the venom gland of *B. arietans*. Its recombinantly expressed form, rBit-3, induced platelet agglutination involving GPIb and VWF and blocked binding of VWF to collagen type III.

In previous studies, bitiscetin-1 and bitiscetin-2 proteins were identified from crude venom powders of supposedly the same *B. arietans* species [18,28]. However, we could not pick up cDNAs coding bitiscetin-1 or -2, and neither did those previous studies find indications of an intra-species family of bitiscetins in the same venom. Our cDNA was derived from a *B. arietans* individual imported from South Africa. The puff adder *B. arietans* inhabits wide regions from the Arabian Peninsula throughout Africa except for deserts and rainforests. The findings of different bitiscetins in this species might be related to the different ages, diets, or habitats of the snakes [45,46], or even to species misidentification [28].

The bitiscetin-3 functions that we identified overlap with those reported for bitiscetin-2 [27,28], and bitiscetin-2 and -3 might represent the same protein. In that case, the sequence differences of bitiscetin-3 with the small sequence fragment determined for bitiscetin-2 (Figure 4A) could be explained by differences between alleles of the same gene. However, that would leave it difficult to explain why neither visualization of bitiscetin-2 after SDS-PAGE under reducing conditions [27] nor N-terminal amino acid sequencing of purified bitiscetin-2 [28] suggested the presence of two different bitiscetin-2 subunits, alpha and beta.

The bitiscetin-1 β subunit sequence was determined by amino acid sequencing [35] as well as cDNA sequencing [39], whereas the bitiscetin-1 α subunit sequence has only been determined at the amino acid level [35]. Comparison of the nucleotide sequences of the bitiscetin-1 and -3 β-subunit open reading frames reveals 90% identity and a dN/dS ratio (a measure for the rate of nonsynonymous versus synonymous substitutions) of 1.5, which implies that their evolutions since separation involved positive selection towards amino acid changes. An accelerated evolution towards a different function is also suggested by bitiscetin-1 and -3 sharing less similarity in the residues that in bitiscetin-1 bind VWF than in their overall sequences (Figure 4A). Future investigation of *B. arietans* genomic sequences should provide more definite answers on the number of bitiscetin genes in this species and the functions of the different bitiscetins.

rBit-3 induced platelet agglutination involving VWF and GPIb, although not as efficiently as bitiscetin-1 (Figure 6A). The involvement of GPIb could be concluded because mAb NMC-4—which binds to the GPIb-binding site on the A1 domain of VWF—blocked rBit-3-induced platelet agglutination (Figure 6B). The involvement of VWF could be concluded because the removal and addition of VWF had a negative and positive effect, respectively, on rBit-3-induced platelet agglutination (Figure 6C). rBit-3 also directly promoted the binding between VWF and GPIb molecules, although not as efficiently as bitiscetin-1 or botrocetin (Figure 7A).

It might also be considered whether bitiscetin-3 function could involve CTL-type lectin activity. Arguing against this is that platelet agglutination induced by rBit-3 does not require the presence of divalent cations, since it was not inhibited by the presence of Ca^2+^-chelating sodium-citrate (0.32%) or 5 mM EDTA in PRP (data not shown). Furthermore, by binding sugars, bitiscetin-3 would lose functional specificity since sugar chains on VWF or GPIb are similar to those carried by other proteins in the plasma. Therefore, we believe that bitiscetin-3 is properly classified as a CTL-like protein and is not a CTL.

Bitiscetin-1 binds to the VWF A1 domain close to the botrocetin-binding site, as determined by mutation assays [21] and X-ray crystallography [25]. Bitiscetin-2 binds to the VWF A3 domain, as has been determined by extensive mutation analyses [28]. For bitiscetin-3, we did not successfully perform direct binding studies (see below), but indirect evidence suggests that it binds to the VWF A3 domain. Namely: (1) A competitive assay for VWF binding between bitiscetin-1 and rBit-3 concluded that their binding site on VWF is different (Figure 7B); (2) rBit-3, but not bitiscetin-1, significantly inhibited the binding between VWF and collagen type III (Figure 8A), and the latter is known to bind VWF A3; (3) VW53-2, an anti-VWF mAb that inhibits VWF binding to collagen type III (and thus probably binds VWF A3) (Figure 8B), inhibited the induction of platelet agglutination by rBit-3 (Figure 8C). Theoretically, these observations might also be explained by rBit-3 directly binding to collagen. However, such a possibility does not agree with the observation that when collagen-coated plates were pre-incubated with rBit-3 solution for 1 h followed by washing, the enhancing effect of rBit-3 was lost, as VWF bound to these plates with a similar efficiency as to plates without preincubation with rBit-3 (data not shown).

We also tried to directly investigate the binding of rBit-3 to VWF by using biotin-labelled rBit-3. Unfortunately, biotin-labelled rBit-3 showed neither VWF-binding nor platelet agglutination inducing activity (data not shown), which we interpreted as the biotinylation of the primary amine groups of rBit-3 affecting its configuration necessary for function. Therefore, we are not showing those results here. In future studies, we will pursue other approaches for studying direct binding of rBit-3 to VWF.

For the binding of VWF to collagen, negatively charged residues in the VWF A3 domain interact with positively charged residues of collagen [47]. The 3D structure of the A3 domain in complex with a collagen type III derived peptide indicates that the binding surface for collagen locates across the VWF A3 domain strand β3 and helices α2 and α3 [48]. Obert et al. [28] introduced Ala mutations in the VWF A3 domain and found that residues Ile^975^, Asp^979^, Pro^981^, Ser^1020^, His^1023^ and Pro^981^ are important for its binding with bitiscetin-2. We have no experimental data about the binding site of bitiscetin on VWF A3, but it might be similar to the bitiscetin-2 binding site.

But if rBit-3 binds to the VWF A3 domain, there is then some question as to how it can stimulate the binding of VWF to GPIb (Figure 7A)—namely, that is a property mediated by VWF A1. Obert et al. [27] found that bitiscetin-2 binding to VWF A3 induced conformational changes in VWF A1 and speculated that these changes enhanced A1 binding to GPIb. However, it may also be worthwhile to consider additional possibilities for how bitiscetin-2 and/or -3 may stimulate VWF-to-GPIb binding. Atomic force microscopy (AFM) studies showed that the VWF A1, A2, and A3 domains can be in different configurations relative to each other, with A1 and A3 in close proximity when in a globular resting state [49,50]. In the model proposed by Crawley et al. [50], this globular state can be stretched out by major changes in the A2 domain configuration, which then exposes the GPIb binding site of A1 which otherwise is shielded from function. Bitiscetin-2 and -3 might destabilize the globular resting state of VWF A1-A2-A3 and so promote exposure of the A1 binding site for GPIb, even without inducing significant changes within the A1 domain. Another possibility is that in VWF bound to GPIb the A1 and A3 domains also may be in close proximity of each other, and that the A3-bound bitiscetin-2 or -3 provides extra surface for binding GPIb. In the case of the latter model, this would be reminiscent of how binding of botrocetin to A1 increases the binding surface for GPIb [26], as also proposed for the binding of bitiscetin-1 to A1 [25]. Several other snake venom proteins also have GPIb-binding activity—Mamushigin from *Agkistrodon halys blomhoffii* [51], alboaggregin-B from *Trimeresurus albolabris* [52], and GPIb-BP from *B. jararaca* [53] are all Snaclec molecules that interact with platelet GPIb, either inducing or inhibiting platelet agglutination. However, other than bitiscetin-2, bitiscetin-3 is the first Snaclec protein known to efficiently block the binding of VWF to collagen type III.

In the future, residues and structures involved in bitiscetin-3 function should be better determined. Since bitiscetin-3 has a relatively poor ability to induce platelet agglutination (Figure 6A), it may be feasible to entirely delete that property. This would create a novel antithrombotic reagent for targeting VWF-collagen interaction. In general, we hope that by modifying and making use of snake venom protein properties, a variety of tools for medical support can be developed. Further investigation is necessary to explore the biological activities and properties of bitiscetin-3.

## 5. Conclusions

In the venom gland of the viper *B. arietans*, we identified transcript sequences for a new Snaclec protein called bitiscetin-3. Like bitiscetin-1 and presumably bitiscetin-2, bitiscetin-3 induces VWF-GPIb-involving platelet agglutination. Bitiscetin-3 and -1 share ~80% amino acid sequence identity, but their binding sites on VWF are very different. Bitiscetin-3 probably binds to the VWF A3 domain. Bitiscetin-3 efficiently inhibits binding between collagen and VWF, as does bitiscetin-2. Bitiscetin-3 is the first Snaclec protein with such a property for which the full α- and β-subunit sequences have been determined, allowing future recombinant studies to address its medicinal potential.

## Figures and Tables

**Figure 1 toxins-14-00236-f001:**
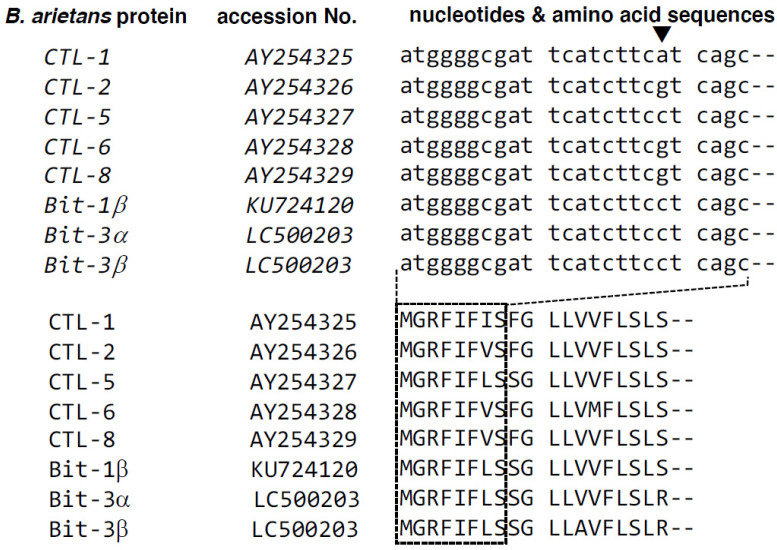
Comparison of signal sequences and their coding sequences of several Snaclec proteins from *B. arietans*. The CTL-1, -2, -5, -6, and -8 sequences were reported by Harrison et al. [38] and bitiscetin-1β by Whiteley et al. [39]. The 24 nt stretch encoding the N-terminal 8 amino acids (lower dotted rectangle) of the signal sequence of these molecules is highly conserved, except for one position (downward triangle). This 24 nt sequence was used to design a degenerate forward primer for PCR. The figure also includes sequences determined in the present study for bitiscetin-3α and -3β.

**Figure 2 toxins-14-00236-f002:**
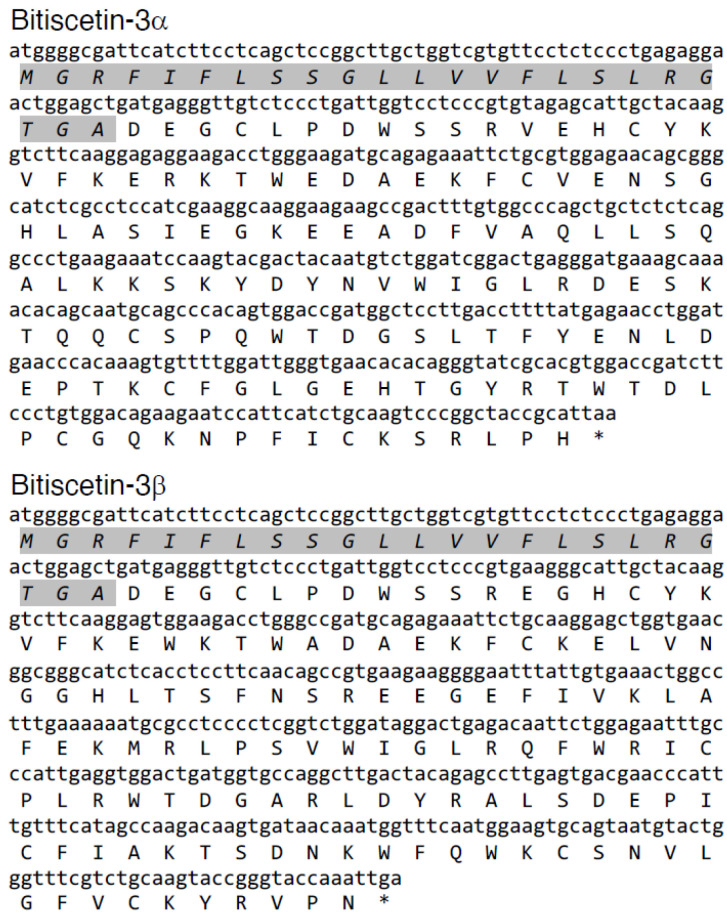
cDNA sequences of bitiscitin-3 subunits. The open reading frames of the α and β subunits of bitiscetin-3 and their encoded amino acid sequences are shown. Both subunits contain signal sequences of 23 amino acid residues (in Italic font plus gray shading).

**Figure 3 toxins-14-00236-f003:**
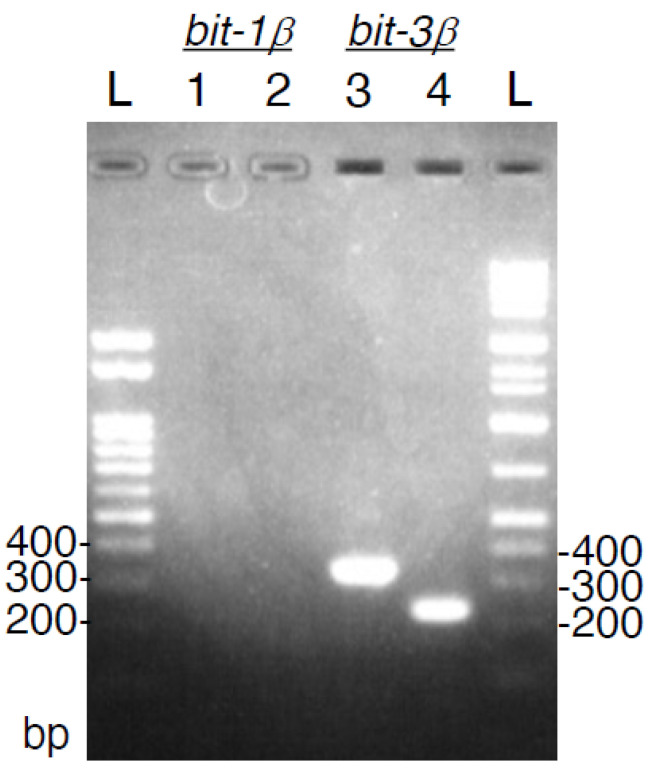
Sequence fragments *of bitiscetin-3β*, but not of *bitiscetin-1β*, could be amplified from the *B. arietans* cDNA sample. Gel electrophoresis of PCR fragments amplified with *bitiscetin-1β* specific primer sets (Lane 1: Bit-1β-F1 + Bit-1β-R1, expected size 329 bp; Lane 2: Bit-1β-F2 + Bit-1β-R12, expected size 234 bp) or with *bitiscetin-3β* specific primer sets (Lane 3: Bit-3β-F1 + Bit-3β-R1, expected size 338 bp; Lane 2: Bit-3β-F2 + Bit-3β-R12, expected size 216 bp). In the right and left lanes, two different ladders are shown.

**Figure 4 toxins-14-00236-f004:**
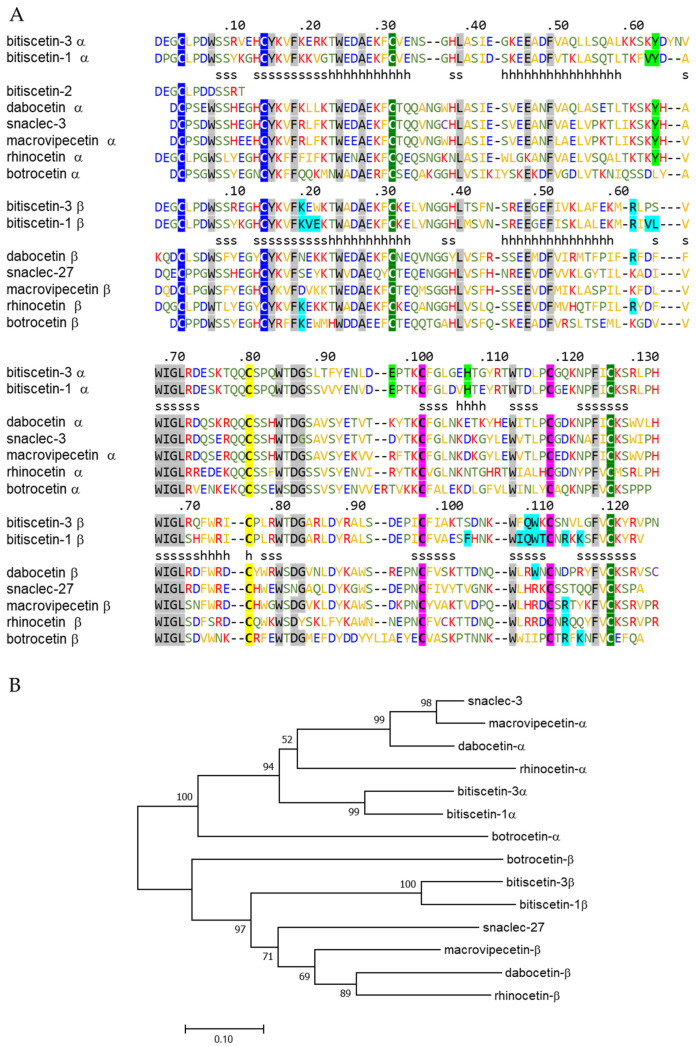
Comparison of amino acid sequences of bitiscitin-3 subunits with other snake venom proteins of the Snaclec family. (**A**) Numbers above the alignment indicate the amino acid numbers of the mature proteins of the respective bitiscetin-3 subunit. The letters h and s indicate helices and β-strands in the bitiscetin-1 structure [25]. Cysteine pairs known or expected to form disulfide bridges (as they are common in CTL proteins) are highlighted by similar color shading. Some other residues that are rather common in CTL proteins [40] are highlighted by gray shading. The characteristic cysteine forming a disulfide bridge between the Snaclec α and β subunits, and which forms part of their swapped regions [14,38], is shaded yellow. Residues that in the bitiscetin-1 plus VWF A1 structure determined by X-ray crystallography [25] are within 4 Å distance of VWF are shaded green and cyan for the α and β subunit, respectively, for bitiscetin-1 as well as for aligned identical residues. Of the non-highlighted residues, the cysteines are depicted in purple, the acidic residues in blue, the basic residues in red, and the green colored residues are less hydrophobic than the orange-colored ones according to Hopp and Woods [41]. Hyphens are used to introduce gaps in the alignment. Dashed lines separate the α and β subunit sequences. Except for the botrocetin sequences, the depicted sequences were chosen, as they showed high similarities with bitiscetin-3 α or β. GenBank accessions for the depicted sequences other than bitiscetin-2 [28] are: bitiscetin-3 α, LC500203; bitiscetin-1 α, Q7LZK5; dabocetin α (from *Daboia siamensis*, Eastern Russel viper), Q38L02; snaclec-3 (from *Vipera ammodytes*, Western sand viper), A0A1I9KNN1; macrovipecetin α (from *Macrovipera lebetina*, Levantine viper), C0HKZ6; rhinocetin α (from *Bitis rhinoceros*, Gabino viper), I7JUQ0; dabocetin α (from *Daboia siamensis*, eastern Russell’s viper), Q38L02; botrocetin α (from *Bothrops jararaca*, jararaca), P22029; bitiscetin-3 β, LC500204; bitiscetin-1β, Q7LZK8; snaclec-27 (from *Echis ocellatus*, West African carpet viper), Q6X5S5; macrovipecetin β (from *M. lebetina*), C0HKZ7; rhinocetin β (from *B. rhinoceros*), I7ICN3; dabocetin β (from *D. siamensis*), Q4PRC6; botrocetin β (from *B. jararaca*), P22030. (**B**) Phylogenetic tree, created by Neighbor-Joining method, of the sequences shown in (**A**). The optimal tree with the sum of branch length = 3.41008546 is shown. The percentage of replicate trees in which the associated taxa clustered together in the bootstrap test (500 replicates) are shown next to the branches if the calculated values were >50%. The tree is drawn to scale, with branch lengths in the same units as those of the evolutionary distances used to infer the phylogenetic tree. The evolutionary distances were computed using the Poisson correction method and are in the units of the number of amino acid substitutions per site.

**Figure 5 toxins-14-00236-f005:**
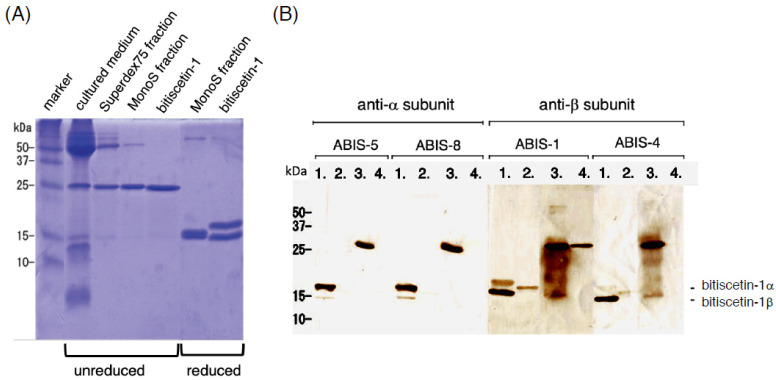
SDS-PAGE of rBit-3 and cross-reactivity to anti-bitiscetin-1 antibodies. (**A**) Aliquots of rBit-3 fractions after various steps of purification, and purified bitiscitin-1 as a control, were subjected to SDS-PAGE. Although the final rBit-3 purification (MonoS fraction) still contains an unknown ~50 kDa band, this preparation was used as rBit-3 in the functional experiments. It shows a major band of ~25 kDa like bitiscetin-1 under nonreducing conditions, but two proximal bands of ~15 kDa, which were distinct from those of bitiscetin-1 under reducing conditions. (**B**) Bitiscetin-1 and rBit-3 were electro-transferred to the PVDF membrane after SDS-PAGE. The membrane was incubated with monoclonal antibodies against bitiscetin-1 subunit α (ABIS-5 and -8) or β (ABIS-1 and 4). lane 1, reduced bitiscetin-1; lane 2, reduced rBit-3; lane 3, unreduced bitiscetin-1; lane 4, unreduced rBit-3. Only ABIS-1 clearly recognized rBit-3.

**Figure 6 toxins-14-00236-f006:**
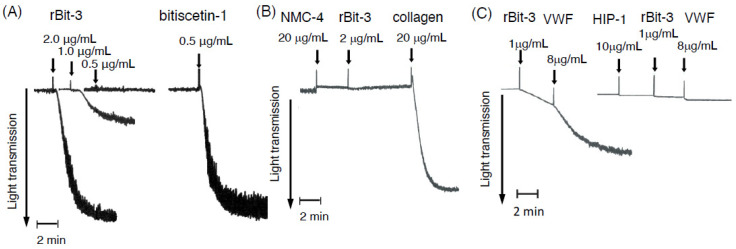
Platelet agglutination induced by rBit-3. The effects of rBit-3 on the platelet-rich plasma (PRP) or washed platelets were examined by light transmission aggregometry. Arrows indicate the timepoints when test solutions were added. (**A**) rBit-3 induced platelet agglutination, but it needed a higher concentration than bitiscetin-1. (**B**) PRP was preincubated with anti-VWF A1 domain monoclonal antibody (NMC-4), after which rBit-3 and collagen were successively added. The preincubation with NMC-4 blocked the agglutination-inducing ability of rBit-3 but not of collagen. (**C**) When washed platelets were used, rBit-3 induced a minor agglutination of platelets, which was accelerated by the subsequent addition of VWF (**left**). The rBit-3-induced platelet agglutination and VWF-induced facilitation were inhibited in the presence of anti-GPIb monoclonal antibody (HIP1) (**right**).

**Figure 7 toxins-14-00236-f007:**
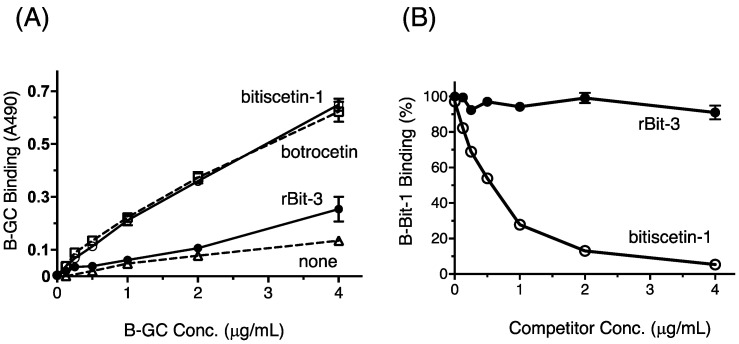
Binding assay of glycocalicin and rBit-3 on VWF using ELISA. (**A**) Binding of biotinylated glycocalicin (B-GC) to the immobilized VWF in the presence of rBit-3 (closed circle), bitiscetin-1 (open circle), botrocetin (square) or TBS (none: triangle) was measured by ELISA. (**B**) Competition assay of rBit-3 and bitiscetin-1 for the VWF binding site. Binding of biotinylated bitiscetin-1 (B-Bit-1: 0.04 µg/mL) to the immobilized VWF in the presence (0–4 µg/mL) of rBit-3 (closed circle) or intact bitiscetin-1 (open circle) was measured by ELISA. The binding in the absence of competitor is assumed as 100% binding. rBit-3 did not inhibit the binding of biotinylated bitiscetin-1 to VWF. Data points indicate the mean ± SE (*n* = 3).

**Figure 8 toxins-14-00236-f008:**
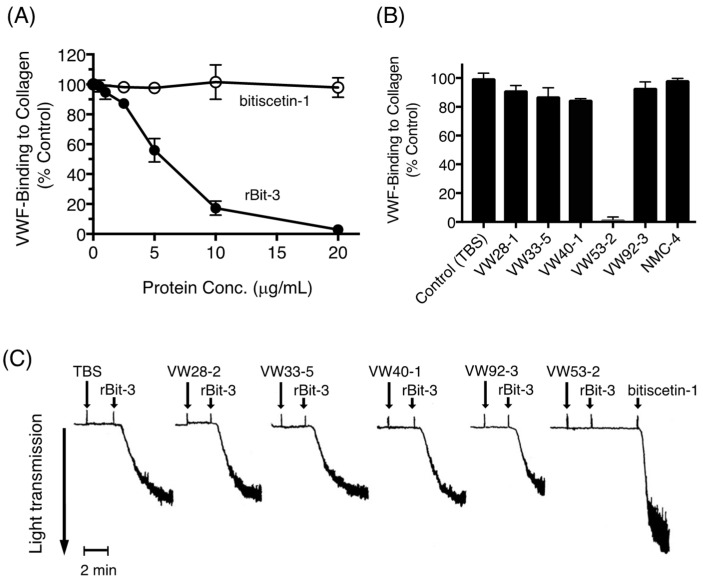
rBit-3 binds to VWF close to the collagen binding site. (**A**) Binding of VWF (0.5 µg/mL) to immobilized collagen was compared in the presence of rBit-3 (closed circle) or bitiscetin-1 (open circle). Data were expressed considering the VWF-binding in the absence of rBit-3 as 100% binding. (*n* = 6) (**B**) Binding of VWF (0.5 µg/mL) to the immobilized collagen was compared in the presence of a monoclonal antibody recognizing a different part of VWF. Data were expressed considering the VWF-binding in the absence of antibody as 100% binding. (*n* = 3) (**C**) Effects of anti-VWF monoclonal antibody (10 µg/mL each) on the platelet agglutination induced by rBit-3 (2 µg/mL) as determined by light transmission aggregometry. Arrows indicate the addition of each test solution.

## Data Availability

Not applicable.

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
