# Peer review of "Bitiscetin-3, a Novel C-Type Lectin-like Protein Cloned from the Venom Gland of the Viper Bitis arietans, Induces Platelet Agglutination and Inhibits Binding of Von Willebrand Factor to Collagen"

_toxins, 2022, doi:10.3390/toxins14040236_

Round 1
Reviewer 1 Report
The structural and functional characterization of new toxins from animal venoms open avenues for future investigations and applications of these biomolecules. In this manuscript, the authors identified and expressed a C-type lectin-like toxin. The study is relevant and enriches the knowledge about the biochemical and structural profile of this class of molecules. Authors should review some points before publication.
- I recommend avoiding abbreviations in the title, even though they are well known in the field.
- Avoid paragraphs of a sentence (lines 50-52, 162-164, 165-166, 168-169, 175-177, 338-339).
- Standardize nomenclature for the purified toxin (lines 15, 81).
- Line 194: NaCl
- Line 204: What blood type? Can it generate different results?
- Please review the order in which the figures are presented and their description in the main text.
- The axes of figures 7 and 8 should end in a number.
Author Response
I have attached the responses to the reviewer 1 as a word file.

Reviewer 2 Report
Reference: Manuscript submitted to TOXINS
General Comments
Reference: Manuscript “Bitiscetin-3, a novel CTL-like protein cloned from the venom 2 gland of the viper Bitis arietans, induces platelet agglutination 3 and inhibits binding of VWF to collagen”.
In the submitted manuscript, the authors present the results of their research with a new toxin isoform belonging to the “Bitiscetin family”, characterized as Bitiscetin-3, a novel C Type Lectin, which was cloned from the venom gland of viper Bitis arietans, and which has aggregating activity of platelets and by binding to von Willebrand Factor inhibits the binding of this extracellular matrix molecule to collagen. The authors co-transfected cDNAs from bitiscetin-3 alpha and beta subunits to 293T cells and obtained the recombinant heterodimer. This purified recombinant protein (rBit-3) was capable to induce platelet agglutination involving VWF and 15 GPIb glycoprotein, did not compete with bitiscetin-1 for binding to VWF, blocked VWF binding to collagen, and inhibited its platelet agglutination ability in the presence of an anti-VWF monoclonal that blocked VWF binding to collagen. After a careful reading of the text, it is my opinion that a minor revised manuscript is suitable and has the quality to be published in Toxins. The subject falls within the scope of the journal, the text is well written, the experiments are relevant, and the group of authors showed knowledge in the area. Some suggestions are attached to improve the quality of the final text.
1- In line 28, of the introduction the authors wrote ..... Toxic proteins in snake venoms can generally be classified as neurotoxins or hemotoxins. It would be better in the corrected version to increase the range of activities of toxins found in snake venoms, as there are dermonecrotic, proteolytic, and ion channel inhibitor toxins, among others.
2- Between lines 31 and 33 the authors wrote ..... A group of snake venom proteins belong to a family of domain-swapped heterodimers of single C-type lectin (CTL)-like domain molecules (“Snaclecs”) [9, 10]. This text is very important because it discusses molecules similar to the one described by them throughout their results, and to me it seemed incomplete. They should write more details of members belonging to this family, to better introduce what to expect from the functional point of view of these molecules!
3- In lines 37-38 the authors wrote .... VWF is a molecular glue that attaches flowing platelets to subendothelial matrices such as collagen at the vascular injury [14,15]. Better replace the word glue by … an extracellular matrix molecule. Although the meaning of the word glue is understood, the term gives a very popular meaning to the text.
4- In figure 2, lines 279-280, where the authors show the amino acid and nucleotide sequences of the alpha and beta subunits of Bitiscetin-3, it would be interesting to emphasize the cysteine ​​residues along the sequence, since they are conserved in the family and surely must be important for molecular functionalities, in addition to confirming that it is the studied molecule.
5- In the text, the figure 4, between lines 283 and 284, appears before figure 3, which appears between lines 315-316. This must be corrected!
6- Also the data in Figure 4, where the authors show an amino acid alignment of Bitiscetin-3, with other members of the family of the Snaclec family, could be used to generate a Cladrogram and thus provide some phylogenetic information, or to determine greater proximity between members!
7- …Upon comparison with protein sequences deposited in the NCBI database, both subunits showed the highest similarity with the respective bitiscetin- 1 subunits, namely 79% and 80% amino acid. Compared to bitiscetin-1, each bitiscetin-3 subunit shows two extra amino acid residues and 27 residue replacements. These data shown between lines 327 – 329 could be reported in a table as results, since they emphasize that the studied molecule is another isoform of the family. An important point of the text!
8- Between lines 331-332, where the authors discuss some mechanistic about the binding of members of the Bitiscetin family to the VWF, this could be shown in a Docking figure, as they have crystallography data! This would surely bring greater attraction to the text. The authors could superimpose the molecules of Bitiscetin-1 and Bitiscetin-3 and discuss their similarities and possibilities in a structural and three-dimensional view. … Several, but not all, of the bitiscetin-1 residues that bind (defined as within 4.0 Å dis- 331 tance) to VWF are conserved in bitiscetin-3 (Fig. 4, light green and cyan shading for bind- 332 -subunit, respectively).
9- In the heterologous expression model used, why did the authors choose to express recombinant Bitiscetin-3 in HEK 293 cells? An embryonic and human lineage? As it is a heterodimer linked by a disulfide bridge, surely the option for bacterial strains would not give a good result, but could not expressions in yeast or insect cells provide a greater protein yield?
10- In figure 5, where the authors show SDS-PAGE and WB of the recombinant samples after expression and purifications of Bitiscetin-3. In the WB experiment using monoclonal antibodies, although the authors state that there was only a positive reaction of Bitiscetin-3 with antibodies that recognize the ABIS-1 domain, it also seems to me that there was a reaction with the antibodies that recognize the ABIS-4 domain, Lane 2 of the figure 5B? What do the authors think of the sign?
11- The text described in lines 351- 353 … N-terminal amino acid 350 sequencing was performed for each of the two bitiscetin-3 bands after their separation Toxins 2021, 13, x FOR PEER REVIEW 10 of 16 sub- b and aunder reducing conditions. The upper and lower band were identified as the unit of bitiscetin-3, respectively. The amino acid sequencing result confirmed that rBit-3 subunits.b and ais a disulfide-linked heterodimer … These amino-terminal sequencing data of the subunits obtained could be part of the figure 5, as an additional table. They are important, confirm previous data and strengthen the results of cloning!
12- In Figure 6C, the authors studied the platelet aggregating activity and suggest that VWF accelerates this aggregating activity. Why they used rBit-3 in the contraction of 1.0 microgram/mL instead of 2.0 microgram/mL, which was the concentration that lead to better platelet aggregation, as shown in Figure 6A? Also what would have happened if they had used other extracellular matrix proteins such as fibrinogen or fibronectin, which are matrix agents known to be involved in platelet aggregation?
13- In the experiments where the authors showed the functionality of rBit-3 in platelet aggregation and the co-participation of the VWF (figures 6 and 7), it would be interesting, and this would bring more attraction to the results, to carry out experiments on adherence of platelet to the VWF, an event that precedes aggregation. Morphological data could bring more learning to the event, mainly cytoskeleton and actin participation.
14- On lines 445-446, the authors wrote … bitiscetin-like protein—bitiscetin-3—in the venom gland of B. arietans. I would replace bitiscentin-like protein with bitiscentin isoform !
15- on line 442. … indications for multiple bitiscetins in the same venom. I suggest to replace by ….indication of an intra-species family of bitiscentins in the same venom.
16- In the line 462 please change SDS-Page by SDS-PAGE!
17- In the line 476 and throughout of the text I suggest replace platelet agglutination by platelet aggregation. To differentiate between aggregation and adhesion!
18- In the lines 503-504 … In future studies, we will pursue other approaches for studying direct binding of rBit-3 to VWF. Authors should be instigated to perform co-crystallization experiments. It is not an easy data to be obtained, but must be tried!
19- Also authors must perform tentative of crystallization of bitiscetin-3. These results will be open an avenue of possibilities for authors!
20- Finally, but not less important, given that Bitiscetin-3 is a C-type lectin, would this recombinant protein have lectin activity (ie recognition of sugar residues)? and would that be important in the functionality of this toxin in the events described? The rBit-3 induced platelet aggregation involving VWF and GPIb is sugar-dependent? That is a good question to be addressed by authors!
Author Response
I have attached the responses to the reviewer 2 as a word file.

Reviewer 3 Report
Minor observations/corrections:
- Abstract, lines 19-23: The last two sentences should be slightly rephrased, as in their current form they can be misconstrued as having bitiscetin-2 sequenced and expressed, instead of bitiscetin-3.
- Introduction, lines 45-47: Similarly, a slight rephrasing would be helpful, as the sentence is hard to interpret.
- Materials and Methods, lines 186 and 192: A space should be introduced between pH and the value (or a “=” symbol).
Questions:
- How many replicates have been used in each stage of the experiment?
- Would it be possible to perform a molecular modelling of VWF A3 domain and bitiscetin-3 binding? It could confirm the indirect evidences described in the Discussion section.
Major concern:
- While the manuscript is well written, the bibliography is poorly selected, from the perspective of novelty. Only about 25% of the references were published in the last 10 years. As numerous methods and protocols are referenced, this is partly understandable. However, the Introduction and Discussion sections are also mostly based on such articles. I propose a review of the recent literature, as many other information could have surfaced in the last decade.
Author Response
I have attached the responses to reviewer 3 as a word file.
